# The Effect of a Combination of *Eucommia ulmoides* and *Achyranthes japonica* on Alleviation of Testosterone Deficiency in Aged Rat Models

**DOI:** 10.3390/nu14163341

**Published:** 2022-08-15

**Authors:** Jeong Yoon Lee, Seokho Kim, Han Ol Kwon, Bong Seok Bae, Sung lye Shim, Woojin Jun, Yoo-Hyun Lee

**Affiliations:** 1Department of Food and Nutrition, The University of Suwon, Hwasung 445743, Korea; 2Korea Ginseng Corporation Research Institute, Korea Ginseng Corporation, Daejeon 34128, Korea; 3Division of Food and Nutrition, Chonnam National University, Gwangju 61187, Korea

**Keywords:** late-onset hypogonadism, testosterone, *Eucommia ulmoides*, *Achyranthes japonica*

## Abstract

With aging, men inevitably encounter irreversible changes, including progressive loss of testosterone and physical strength, and increased fat mass. To assess the alleviatory effects of EUAJ on andropause symptoms, including in vivo testosterone deficiency, we administered EUAJ for 6 weeks in 22-week-old Sprague-Dawley rats. Before EUAJ (3:1) (*E. ulmoides*:*A. japonica* = 3:1, KGC08EA) administration, testosterone decline in 22-week-old SD rats was confirmed compared to 7-week-old SD rats (NC group). After administration of EUAJ (3:1) at 20, 40, and 80 mg/kg for 6 weeks, testosterone, free testosterone, and mRNA expression levels (*Cyp11a1* and *Hsd3b1*) were significantly increased at 40 mg/kg EUAJ (3:1), whereas mRNA expression levels of *Cyp19a1* and *Srd5a2* were significantly reduced at this concentration, compared to the control group. Swimming retention time was significantly increased at both 40 mg/kg and 80 mg/kg. In summary, EUAJ (3:1) enhanced testosterone production by increasing bioavailable testosterone, sex hormone-binding globulin (SHBG), and enzymes related to testosterone synthesis at 40 mg/kg. In addition, 80 mg/kg EUAJ (3:1) also increased physical and testicular functions.

## 1. Introduction

Androgen deficiency in aging males (ADAM) is the main characteristic of andropause, age-related physiological changes associated with low testosterone (T) levels [1]. Testosterone decreases by approximately 1% every year from the age of 30 [1]. Total serum T concentration under 8 nmol/L (231 ng/dL) causes late-onset hypogonadism (LOH), the prevalence of which is less than 20% until a male reaches 60-years-old, but doubles after a male reaches 70 [2]. Thus, andropause is usually accompanied by typical symptoms of aging, with LOH symptoms due to low serum testosterone levels. The main symptoms of andropause are similar to those of female menopause, such as reduced bone densitometry, sleep disorder, increased body fat, depression, and fatigue, and include LOH symptoms, such as decreased libido, erectile dysfunction, and loss of body hair [3]. Andropause not only reduces the quality of life of elderly men but is also highly associated with mortality [4].

Testosterone synthesis occurs mainly in the testicular Leydig cells. Approximately 98% of serum T is in the form of protein-bound T, such as sex hormone-binding globulin (SHBG) [5]. As aging progresses, SHBG-bound T increases and free T decreases [6]. In general, serum T and SHBG levels are used as biochemical tests for LOH. Treatment is carried out when total T concentration is less than 8 nmol/L (231 ng/dL) or when free T concentration is less than 180 pmol/L (52 pg/mL). Testosterone replacement therapy (TRT) is a common treatment for LOH. However, it has limitations due to long-term effects, safety, and application [7]. The increase in life expectancy has raised the desire to improve quality of life in old age. Therefore, the demand for functional foods for LOH using safe natural products, with minimal application limitations, has also increased. *Eucommia ulmoides* and *Achyranthes japonica* are traditional medicinal plants common in Asia. Both strengthen bones and muscles and are used in botanical nourishing tonics. *E. ulmoides* improves diabetes related to hypogonadism and inhibits the growth of osteoclasts in senescence-accelerated mouse P6 [8,9]. Various outcomes, such as in vitro and in vivo anti-inflammatory effects and improvement in bone-related functions in ovariectomized mice, have been reported [10,11]. In a previous experiment, we confirmed in vitro that the *E. ulmoides* and *A. japonica* complex increased steroidogenic enzymes that catalyze T synthesis and T levels in TM3 Leydig cells. In this study, the effects of *E. ulmoides* and *A. japonica* on andropause-induced Sprague-Dawley rats were investigated. Our findings might aid in the development of an effective treatment strategy for LOH.

## 2. Materials and Methods

### 2.1. Materials

The bark of *E. ulmoides* and radix of *A. japonica* were purchased from Kyungil Medicinal Herbs (Geumsan, Korea), and subsequently mixed (EUAJ = *E. ulmoides*:*A. japonica*) and extracted twice with 30% ethanol. The extracts were filtered, concentrated under a vacuum, and spray-dried (with yields of 11.0% and 45%, respectively). All distilled water used in the experiment was purified by Milli-Q water system (Millipore, Bedford, MA, USA), and resistance was measured as 18 MΩ prior to use. All reagents used in the experiments were of guaranteed reagent grade, and HPLC-grade acetonitrile and methanol were purchased from Merck (Darmstadt, Germany). The standard reference materials pinoresinol diglucoside (purity: 98.9%) and 20-hydroxyecdysone (purity: 98.0%) were purchased from Chromadex and Sigma-Aldrich (St. Louis, MO, USA), respectively.

### 2.2. Sample Preparation and Liquid Chromatography

EUAJ (3:1) (*E. ulmoides*:*A. japonica* = 3:1, KGC08EA) (200 mg) was weighed in a volumetric flask, and 70% MeOH was added to analyze pinoresinol diglucoside and 20-hydroxyecdysone contents. Each solution was filtered through a 0.2 μm filter (PVDF, Whatman, UK) and injected into an HPLC system (high performance liquid chromatography, Waters Co., Milford, MA, USA). Chromatographic separation for pinoresinol diglucoside was performed using a Supelco Discovery C18 column (5.0 μm, 4.6 × 250 mm). The column temperature was maintained at 25 °C. The detection was monitored at a wavelength of 230 nm. The mobile phase consisted of acetonitrile (ACN; solvent A) and 0.05% phosphate buffer (solvent B) at a flow rate of 1.0 mL/min. The gradient conditions were 0–5 min (5% A), 20 min (20% A), 30 min (30% A), 35 min (95% A), 35–36 min (95% A), 36–40 min (5% A), adjusted sequentially. A Halo C18 column (2.7 μm, 4.6 × 150 mm) was used for 20-hydroxyecdysone contents analysis. The detection wavelength was 250 nm. The column and autosampler tray temperatures was at 35 °C. The mobile phase consisted of acetonitrile (ACN; solvent A) and distilled water (solvent B) at a flow rate of 0.7 mL/min. The conditions of solvent gradient were 0–1 min (10% A), 61 min (13% A), 62 min (90% A), 62~66.1 min (90% A), and 67 min (10% A) 67–70 min (10% A).

### 2.3. Experimental Animals

The 12-week-old Sprague-Dawley rats (300–350 g) were purchased from the Central Animal Lab Inc. (Seoul, Korea). The rats were aged naturally for 10 weeks; 22-week-old rats (500–550 g) were used as the andropause animal model and 7-week-old SD rats (200–250 g) were used as a non-andropause model. For exercise capacity evaluation, ICR mice were purchased and forced to swim. All animal experiments were approved by the IACUC (Institutional Animal Care and Use Committee) of Suwon University (approval number: USW-IACUC-2020-001). Animals were housed in a room with light controlled at 12 h intervals, maintained at 22 ± 2 °C, and a relative humidity of 55 ± 5%. They were fed normal chow (AIN-76) and allowed to drink tap water at any time. Before the experiment, orbital blood was collected to measure serum testosterone levels. We confirmed that the testosterone concentration in 22-week-old rats (andropause model) was significantly lower than that in 7-week-old rats (non-andropause model). To screen the extracts, SD rats were administered various mixture ratios of *A. japonica* (aj) and *E. ulmoides* (eu). There were a total of 7 groups, which were as follows: control; 22-week-old SD rats orally administered saline, aj; 22-week-old SD rats orally administered aj at a concentration of 40 mg/kg, eu; SD rats orally administered eu at a concentration of 40 mg/kg, (3:1); 22-week-old SD rats orally administered aj and eu at a concentration of 40 mg/kg each at a ratio of 3:1, (1:1); 22-week-old SD rats orally administered aj and eu at a concentration of 40 mg/kg each at a ratio of 1:1, (1:3); 22-week-old SD rats orally administered aj and eu at a concentration of 40 mg/kg at a ratio of 1:3, NC; 7-week-old SD rats orally administered saline. Six weeks later, animal blood was collected again to measure serum testosterone concentration. The mixture ratio of aj and eu, which demonstrated the greatest increase in testosterone levels, was selected and named EUAJ.

EUAJ was evaluated again at various concentrations. Newly purchased SD rats were administered 20, 40, and 80 mg/kg EUAJ for 6 weeks. There was a total of five groups, which were as follows: control; 22-week-old SD rats orally administered saline, low; 22-week-old SD rats orally administered EUAJ at a concentration of 20 mg/kg, middle; 22-week-old SD rats orally administered EUAJ at a concentration of 40 mg/kg, high; 22-week-old SD rats orally administered EUAJ at a concentration of 80 mg/kg, NC; 7-week-old SD rats orally administered saline. Thereafter, T levels, free T levels, ALT, AST, lipid profile, SHBG, estradiol levels, and PSA levels in SD rat serum were investigated. Steroidogenesis gene mRNA expression was also confirmed in animal testis tissue. Finally, epididymal fat ratio, muscle weight ratio, and sperm count were investigated.

We evaluated the effect of EUAJ on exercise capacity in ICR mice. Naturally aged 22-week-old ICR mice were also administered 20, 40, and 80 mg/kg EUAJ and performed swimming exercises. In addition, 7-week-old ICR mice were used as positive controls and not subjected to exercise. There were a total of 5 groups, which were as follows: control; 22-week-old SD rats orally administered saline with exercise, low; 22-week-old ICR mice orally administered EUAJ at a concentration of 20 mg/kg with exercise, middle; 22-week-old ICR mice orally administered EUAJ at a concentration of 40 mg/kg with exercise, high; 22-week-old ICR mice orally administered EUAJ at a concentration of 80 mg/kg with exercise, NC; 7-week-old ICR mice orally administered saline. Then, T levels, malondialdehyde (MDA), and lactate dehydrogenase (LDH) were confirmed in ICR mouse serum. In addition, the presence of muscle glycogen was confirmed in animal muscle tissue, and swimming retention time was investigated for a total of 4 weeks. All blood collection and oral administration was performed in the morning (9–11 am).

### 2.4. Serum Testosterone Concentration Assay

Animal blood samples were collected and centrifuged for 20 min at 1500× *g* at room temperature. Serum testosterone concentrations were measured using a Testosterone ELISA Kit (Abcam, Cambridge, UK) and the DetectX^®^ Testosterone ELISA Kits (Arbor assays, MI, USA). Free testosterone was measured using the Rat Free Testosterone (F-TESTO) ELISA Kit (Cusabio, Wuhan, China). The absorbance was measured at 450 nm using a microplate spectrophotometer. All data were calculated using the Curve Expert 1.3 (Hyams Development).

### 2.5. SHBG Levels in Serum

To investigate the correlation between SHBG binding capacity and age-related testosterone deficiency, serum-bound SHBG levels in experimental animals were measured using the Rat SHBG ELISA Kit (Elabscience, Houston, TX, USA), according to the manufacturer’s instructions. The absorbance values measured at 450 nm using a microplate spectrophotometer were numerically calculated and compared using the Curve Expert 1.3 (Hyams Development).

### 2.6. Serum Estradiol Levels Measurement

To determine whether a disproportion between sex hormones can affect male reproductive function, serum estradiol levels of experimental animals were measured using a mouse/rat estradiol ELISA kit (Calbiotech, Mannheim, Germany), according to the manufacturer’s instructions. The absorbance values measured at 450 nm using a microplate spectrophotometer were numerically calculated and compared using the Curve Expert 1.3 (Hyams Development).

### 2.7. DNA Synthesis and Quantitative Real-Time PCR

Quantitative real-time PCR was performed to compare the expression levels of androgen synthesis-related enzymes. After the testis tissue of each experimental animal was collected and homogenized, RNA was isolated using RNAiso Plus (Takara, Shiga, Japan). After synthesizing the same amount of RNA as cDNA using reverse transcriptase and measuring the concentration and purity at 260/280 nm absorbance, real-time PCR was performed using a Roche Lightcycler 96 Real-time PCR System (Roche, Basel, Switzerland). The primers used were β-actin, *Cyp11a1*, *Hsd3b1*, *Hsd17b3*, *Srd5a2* and *Cyp19a1*. The sequences of each primer are shown in Appendix A.

### 2.8. ALT, AST Analysis in the Serum

To evaluate the safety of the functional materials, hepatotoxicity tests were performed to measure the levels of alanine aminotransferase (ALT) and aspartate aminotransferase (AST) in serum. The blood of the experimental animals was centrifuged at 3000 rpm, and only the serum was collected and used for analysis. The GOT·GPT measurement solution (Asan Pharm. Co., Ltd., Hwaseong-si, Korea) was used for analyses. The values measured at 505 nm were converted into Karmen unit (KU) and compared.

### 2.9. Biochemical Analysis

Biochemical analyses were performed to investigate the changes in epididymal fat, muscle mass, lipid profile, muscle glycogen, lactate dehydrogenase (LDH), and malondialdehyde (MDA), which are associated with aging-related testosterone decline. The blood of experimental animals was centrifuged at 3000 rpm for 20 min, and only serum was collected for biochemical analysis. The muscles (soleus, plantaris, EDL, tibialis anterior, and gastrocnemius muscle) and epididymal fat were also extracted and weighed. Triglycerides (TG), total cholesterol, HDL-cholesterol, and LDL-cholesterol levels were measured using a chemistry analyzer (FUJI DRICHEM 4000; Fujifilm, Tokyo, Japan). Lactic dehydrogenase (LDH), malondialdehyde (MDA), and muscle glycogen levels were measured using commercially available kits from Sigma-Aldrich (St. Louis, MO, USA).

### 2.10. Sperm Count Measurement

After sacrificing the experimental animals, the testes were removed, placed in a container containing Hank’s balanced salt solution (HBSS) buffer, finely chopped, and homogenized. The homogenized solution (500 μL) was diluted in 10% formaldehyde fixative. After collecting 10 μL of the diluted solution and leaving it for 7 min in a hemocytometer, the sperm count was measured using an optical microscope.

### 2.11. Swimming Retention Time of Experimental Animals

To measure the animal’s swimming retention time, the forced swimming test (FST) was performed once a week, using a weight equivalent to 5% of its body weight hung on its tail using a plastic barrel, filled with approximately 100 L of water. The indoor temperature was 25 °C and the water temperature was maintained at 30–35 °C inside a simple swimming pool. The swimming retention time of the animal was considered until exhaustion, when it did not come up without external help from the sinking state for approximately 10 s, and swimming was stopped.

### 2.12. PSA Levels in Serum

To evaluate safety against testosterone increase, serum PSA level of the experimental animals was measured using a Rat Prostate Specific Antigen (PSA) ELISA kit (Cusabio, Wuhan, China) according to the manufacturer’s instructions. The absorbance values measured at 450 nm using a microplate spectrophotometer were numerically calculated and compared using the Curve Expert 1.3 (Hyams Development).

### 2.13. Statistical Analysis

All data are expressed as the mean ± standard deviation (SD) of at least triplicate experiments. Statistical significance was analyzed by one-way ANOVA and Duncan’s multiple range test using SPSS Statistics 22 software (International Business Machines Corp., Armonk, NY, USA). Statistical significance was set at *p* < 0.05.

## 3. Results

### 3.1. Effects of EUAJ (3:1) on Testosterone Concentration in Serum of Aged SD Rats

Figure 1A shows that the testosterone levels in 22-week-old SD rats were significantly lower than those in 7-week-old SD rats by more than 50% (*p* < 0.05) before the experiment. Six weeks after the administration of each sample (40 mg/kg body weight), the EUAJ (3:1) group (121.71 ± 131.08%) displayed a significant increase of approximately 7.5 times compared to the control group (*p* < 0.05) (Figure 1B). EUAJ (3:1) was selected as the final sample for this experiment.

### 3.2. HPLC Analysis of Pinoresinol Diglucoside and 20-Hydroxyecdysone in EUAJ (3:1)

A proprietary blend of EUAJ (3:1) is a mixture of *Eucommia ulmoides* and *Achyranthes japonica* 30% ethanol extract at a ratio of 3:1. The contents of the two ingredients were 20.09 mg/g pinoresinol diglucoside from *Eucommia ulmoides* and 0.41 mg/g 20-hydroxyecdysone from *Achyranthes japonica* (Figure 2).

### 3.3. Effects of EUAJ (3:1) at the Various Concentrations of T, SHBG and Estradiol in Serum of Aged SD Rats

Figure 3A,B show the differences in testosterone levels after 6-week administration of EUAJ (3:1) in aged rats. The testosterone level of the middle EUAJ (3:1) group (101.91 ± 48.47%) was significantly increased about 2.6 times compared to the control group (*p* < 0.05). The free testosterone level of the middle EUAJ (3:1) group (17.13 ± 3.08 pg/mL) was significantly increased about 2.4 times compared to the control group (*p* < 0.05) (Figure 3C). The SHBG level of the middle EUAJ (3:1) group was not significantly different from that of the NC group, whereas the level in the control group was approximately 1.7 times higher than that in the middle EUAJ (3:1) group (Figure 3D). The estradiol level of the control group (244.17 ± 21.28 pg/mL) was about 1.3 times higher than the NC group (185.56 ± 4.09 pg/mL). On the other hand, by EUAJ administration, the estradiol level was significantly reduced in all EUAJ groups, similar to the NC group (*p* < 0.05) (Figure 3E).

### 3.4. Effects of EUAJ (3:1) on fertility

The sperm count of the SD rats was examined and is shown in Figure 4. Sperm count of the control group (8.63 ± 1.46 × 10^6^/mL) was approximately 1.6 times lower than that of the NC group (13.58 ± 0.54 × 10^6^/mL). However, the level in the high EUAJ (3:1) group significantly increased compared to the NC group (*p* < 0.05).

### 3.5. EUAJ (3:1) Stimulates Testosterone Synthesis Pathway by Upregulating Expression of Steroidogenic Genes

To assess the effects of EUAJ (3:1) on the mRNA expression of genes related to testosterone synthesis, quantitative real-time PCR was conducted (Figure 5). The enzymes in the steroidogenic pathway, such as cholesterol side chain cleavage enzyme (P450scc), 3β-HSD, and 17β-HSD, catalyze testosterone synthesis, in contrast to aromatase and 5α-reductase2. *Cyp11a1* (P450scc) mRNA expression was significantly increased by 2.03 times for middle EUAJ (3:1) and 2.2 times for high EUAJ (3:1), compared to that in the control group (*p* < 0.05) (Figure 5A). *Hsd3b1* (3β-HSD) mRNA expression in the middle EUAJ (3:1) group significantly increased by approximately 1.8 times compared to that in the control group (*p* < 0.05) (Figure 5B). The mRNA expression of *Hsd17b3* (17β-HSD) in the EUAJ (3:1)-administered groups increased in a dose-dependent manner (Figure 5C). *Cyp19a1* (aromatase) and *Srd5a2* (5α-reductase2) mRNA expression was significantly reduced by administration of EUAJ (3:1) compared to that in the NC group (*p* < 0.05) (Figure 5D,E).

### 3.6. ALT and AST Levels

The liver toxicity of EUAJ (3:1) was evaluated by measuring liver weight and serum alanine aminotransferase (ALT) and aspartate aminotransferase (AST) levels in SD rats (Table 1). The liver weight ratio, serum ALT, and AST of the groups administered EUAJ (3:1) did not increase compared to those of the control group (*p* < 0.05). These results demonstrated that EUAJ (3:1) did not exhibit hepatotoxicity in experimental animals at high concentrations (80 mg/kg).

### 3.7. Effects of EUAJ (3:1) on Body Composition and Serum Lipid Profiles in Aged SD Rats

Changes in epididymal fat, muscle mass, and serum lipid profiles in aged SD rats administered EUAJ (3:1) were measured (Table 2 and Table 3). The epididymal fat weight in the control group was approximately 1.9 times that in the NC group (Table 2). The middle EUAJ (3:1) group (8.68 ± 1.03 g) significantly decreased to approximately 22.7% of the control group (11.24 ± 0.44 g) (*p* < 0.05). In addition, the decrease in epididymal fat weight ratio in the middle EUAJ (3:1) group was significantly reduced, by approximately 22.97%, compared to the control group. The muscle mass of the control group was approximately 12.9% lower than that of the NC group (*p* < 0.05). However, the middle and high EUAJ (3:1) groups showed significant increases in muscle mass of 9.41% and 10.11%, respectively (*p* < 0.05). The lipid profiles of the SD rats are shown in Table 3. The serum TG level in the middle EUAJ (3:1) group was significantly lower than that in the control group (*p* < 0.05). The higher the concentration of EUAJ (3:1), the greater the decrease in TG level.

### 3.8. EUAJ (3:1) Enhances Physical Function and Stamina through Testosterone Elevation in ICR Mice

As shown in Figure 6A, serum testosterone levels in ICR mice orally administered 40 mg/kg EUAJ (3:1) significantly increased (*p* < 0.05). To evaluate physical function in aged ICR mice after administration of EUAJ (3:1), swimming retention time was measured by a forced swimming test once a week (Figure 6B). Administration of EUAJ (3:1) increased swimming retention time by up to 2-fold at medium concentrations (40 mg/kg) and 1.8 times at high concentrations (80 mg/kg), with an increase in testosterone levels (*p* < 0.05). Before sacrifice, the mice were forced to swim for 30 min with loads and then removed from each group for analysis of muscle glycogen and blood biochemical parameters (Figure 6C–E). Muscle glycogen in the middle and high EUAJ (3:1) groups showed approximately 122.13% and 145.11% steep increases, respectively, compared to the control group, which was similar to the NC group (*p* < 0.05) (Figure 6C). The MDA level of the control group was 36.62% higher than that of the NC group, and all EUAJ (3:1) groups, except the middle group, recorded a lower MDA level than the control group (*p* < 0.05) (Figure 6D). LDH level in the control group was approximately twice as high as that in the NC group (*p* < 0.05). Administration of EUAJ at a concentration of 80 mg/kg significantly reduced LDH by 50.56% compared to that of the control group, which was similar to the NC group (*p* < 0.05) (Figure 6E).

### 3.9. PSA Level According to the Administration of EUAJ (3:1) in SD Rats

The PSA level according to the administration of EUAJ (3:1) was measured in the serum of SD rats by ELISA (Figure 7). There was no significant difference in PSA levels between the groups. These results showed that EUAJ (3:1) did not present a risk for prostate disease in experimental animals up to high concentrations (80 mg/kg).

## 4. Discussion

Late-onset hypogonadism (LOH) is accompanied by low testosterone levels and aging-related clinical symptoms, such as sexual function, fat mass, muscle mass, and bone density [12]. A previous in vitro study showed that EUAJ (3:1) increased testosterone levels. Therefore, we investigated the effects of EUAJ (3:1) on LOH in aged SD rats and ICR mice.

EUAJ (3:1) is a mixture of *Encommia ulmoides* (eu) and *Achyranthes japonica* (aj) 30% ethanol extracts at a ratio of 3:1 and contains pinoresinol diglucoside (PDG) and 20-hydroxyecdysone (20HE) as active ingredients. PDG is a lignan mainly present in the bark of *Eucommia ulmoides* [13] and 20HE, present in *Achyranthes japonica*, is an ecdysteroid synthesized by plants [14].

In the blood, testosterone is bound to sex hormone-binding globulin (SHBG) or albumin, or exists as free testosterone. SHBG does not affect free testosterone levels; however, aged men have increased SHBG levels. Testosterone bound to SHBG is difficult to obtain in vivo due to its strong binding power [15,16]. In this study, medium EUAJ (3:1) reduced SHBG and increased free testosterone, suggesting an increase in bioavailable testosterone. Testosterone is produced in the Leydig cells of the testis, and enzymes involved in synthesis include cholesterol side chain cleavage enzyme (P450scc), 3β-HSD, and 17β-HSD [17]. Specifically, testosterone synthesis is initiated by the binding of LH secreted by the pituitary gland to the LH receptors of Leydig cells [18]. After the LH reaction, steroid acute regulatory (StAR) protein, expressed through the cAMP pathway, moves cholesterol outside the mitochondria to the inner mitochondrial membrane and is converted to pregnenolone by P450scc [19]. Among the enzymes involved in testosterone synthesis, 3β-HSD converts pregnenolone, DHEA, and androstenediol to progesterone, androstenedione, and testosterone, respectively. 17β-HSD converts DHEA and androstenedione into androstenediol and testosterone, respectively [20]. In contrast, estradiol is converted from testosterone by aromatase [21], and DHT converted from testosterone by 5α-reductase2 causes prostate disease [22]. In relation to sex hormones, an important diagnostic criterion for LOH, middle EUAJ (3:1) displayed increased testosterone levels and decreased estradiol levels. This is due to the increased mRNA expression of *Cyp11a1* (P450scc) and *Hsd3b1* (3β-HSD), which catalyze the testosterone synthesis pathway, while reducing the mRNA expression of *Cyp19a1* (aromatase) and *Srd5a2* (5α-reductase2), which degrade testosterone.

In the steroidogenesis pathway, the preference for either Δ4 or Δ5 pathways in species differences is due to relative substrate affinity of the CYP17 enzyme. In the rat model, CYP17 shows a preference for the Δ4 pathway, containing the intermediates progesterone and 17α-hydroxy-progesterone [23]. In this study, *Hsd3b1* showed a significant increase at the middle concentration of EUAJ (3:1), and it means that the role of *Hsd3b1* in the pathway from Δ5 to Δ4 might be relatively increased at the middle concentration. This result led to a predominant increase in testosterone at the middle concentration of EUAJ (3:1).

Patients can undergo testosterone replacement therapy to increase their testosterone levels. However, this can also lead to increased PSA levels, which are associated with prostate cancer; therefore, periodic monitoring is required [24]. There were no significant differences in serum PSA levels between the groups with or without administration of EUAJ (3:1).

In addition, there was no toxicity in rats at concentrations up to 11.2 g/kg when *Eucommia ulmoides* extract was orally administered for 13 weeks [25] and up to 2 g/kg when *Achyranthes japonica* extract was orally administered for 4 weeks [26]. In rats orally administered EUAJ (3:1) for six weeks, there was no change in serum ALT and AST levels. At high concentrations, ALT decreased, which is thought to be a result of the hepatoprotective effect of *E. ulmoides* [27,28,29,30].

Increased fat mass is a clinical symptom of LOH and is associated with decreased testosterone levels [31]. Obesity and aging can lead to LOH. Conversely, low testosterone levels increase body fat and changes in body composition heighten the risk of obesity and diabetes [12]. Studies have shown that decreased testosterone levels in men result in increased body fat and reduced insulin sensitivity [31,32]. Middle EUAJ (3:1) may have a positive effect on changes in body composition, such as reducing epididymal fat and increasing muscle mass; however, additional research is needed to verify this.

Blood lipid profiles are associated with LOH, and TG, total cholesterol, and LDL cholesterol levels increase as testosterone levels decrease [33]. Low testosterone significantly increased the TG/HDL-cholesterol ratio; testosterone and the TG/HDL-cholesterol ratio are related [34]. In this study, EUAJ (3:1) did not affect the increase or decrease in HDL cholesterol, but decreased TG levels. The increase in testosterone due to administration of EUAJ (3:1) may contribute to the decrease in TG levels.

Testosterone, FSH, and LH affect spermatogenesis [35]. The decrease in number of Leydig and Sertoli cells in the testes due to aging leads to lower sperm production [36,37]. Middle EUAJ (3:1) significantly increased testosterone compared to high EUAJ (3:1) and high EUAJ (3:1) significantly increased sperm count compared to middle EUAJ (3:1) (Figure 4). Although testosterone may affect spermatogenesis, spermatogenesis may also increase by other factors*. Eucommia ulmoides* has been used as a traditional medicinal plant supplement to prevent male infertility. According to a recent study, the Chinese prescription, Duzhong Butiansu (DZBTS), which contains *E. ulmoides* Oliv., down-regulates heat stress, thereby reducing oxidative stress and improving spermatogenesis [38]. In addition, aucubin, an iridoid glucoside extracted from *E. ulmoides*, protected spermatogenesis by preventing apoptosis of sertoli cells by upregulation of Nrf2 and induction of antioxidant responses from triptolide-induced testicular damage [39].

Testosterone affects muscle mass, bone mass, and physical functions [40,41]. Testosterone has a strong anabolic effect on skeletal muscle, regulates protein metabolism, and is closely related to muscle protein synthesis [42,43,44,45,46]. The increase in testosterone, resulting from treatment with EUAJ, appears to lead to an increase in muscle mass. One study demonstrated decreased physical activity in SHBG transgenic mice and increased physical activity in AR-knockout mice [47]. This indicates that testosterone, especially bioavailable testosterone, can increase physical activity. Similarly, middle EUAJ (3:1) increased testosterone levels in rats and mice, and swimming retention times of the experimental animals were consequently elevated.

EUAJ (3:1) also displayed anti-fatigue effects by increasing muscle mass and glycogen and decreasing serum LDH and MDA levels. Lactate is produced during glycolysis and is an important indicator of exercise fatigue [48]. Lactic dehydrogenase (LDH), a cytosolic enzyme, is physiologically abundant in the cells of organs, including the heart, kidney, and skeletal muscles. Vigorous exercise fatigue results in the release of reactive oxygen species (ROS) that cause lipid peroxidation of membrane structures and lead to damage of myocytes and leakage of LDH into plasma [49]. Lipid peroxidation directly contributes to the pathophysiology of fatigue [49]. Under fatigue conditions, MDA, the main product of membrane lipid peroxidation by free radicals, increases and directly contributes to the pathophysiology of fatigue [48,50]. EUAJ (3:1) ameliorated age-related muscle loss by increasing muscle mass and improving aging-related fatigue by raising muscle glycogen and decreasing serum LDH and MDA levels.

## 5. Conclusions

In conclusion, EUAJ (3:1) elevated bioavailable testosterone by increasing total and free testosterone levels, while decreasing SHBG and estradiol levels. Testosterone increase appears to be due to the upregulation of mRNA expression of steroidogenic genes, such as *Cyp11a1* (P450scc), *Hsd3b1* (3β-HSD), and *Hsd17b3* (17β-HSD), which are related to testosterone synthesis following EUAJ (3:1) administration. With heightened testosterone levels, EUAJ (3:1) improved fertility by increasing sperm count and enhanced physical activity through delayed swimming retention time. EUAJ (3:1) also reduced epididymal fat and serum triglyceride levels, while increasing muscle mass and alleviating changes in body composition caused by aging. Furthermore, EUAJ (3:1) exhibited the potential for fatigue improvement by increasing muscle glycogen and decreasing MDA and LDA.

Our findings suggest that EUAJ (3:1) ameliorated andropause symptoms by increasing testosterone levels and enhancing physical function (Figure 8). Similar to a previous study [51], EUAJ (3:1) restored the age-related decline in testosterone in aged SD rats by modulating steroidogenesis gene expression in the testes. Bioavailable testosterone is a decisive factor in muscle strength and physical function. Over 60% of sex hormones bind to SHBG, which increases inversely with obesity or hyperinsulinemia; the opposite is true in the case of aging [52]. Body fat mass of middle-aged and elderly men is strongly associated with low sex hormone levels [53]. Therefore, it appears that serum TG and epididymal fat decrease with EUAJ (3:1) treatment, leading to a reduction in SHBG (Figure 8). An increase in bioavailable testosterone has a positive effect on muscle mass, as well as muscle glycogen and biochemical parameters, such as LDH and MDA, leading to improved athletic performance and stamina. Taken together, EUAJ (3:1) may be employed in future studies for its potential in the treatment of andropause.

## Figures and Tables

**Figure 1 nutrients-14-03341-f001:**
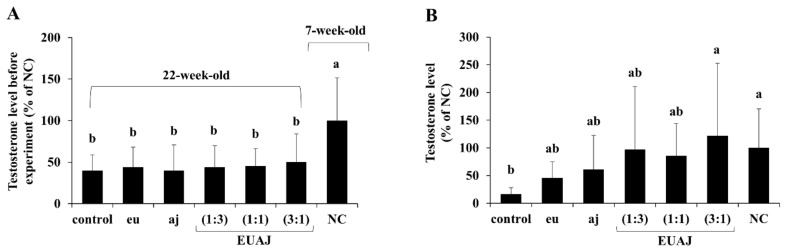
The Effects of EUAJ on testosterone levels in serum of SD rats. Serum testosterone levels of SD rats administered by eu, aj and its combinations in various proportions were shown in (**A**) and (**B**). Values are presented as means ± SD. Different superscript letters show a significantly difference at *p* < 0.05 as determined by Duncan’s multiple range test. Control; saline, aj:aj (40 mg/kg), eu:eu (40 mg/kg), EUAJ (1:3); eu:aj = 1:3 (40 mg/kg), EUAJ (1:1); eu:aj = 1:1 (40 mg/kg), EUAJ (3:1); eu:aj = 3:1 (40 mg/kg).

**Figure 2 nutrients-14-03341-f002:**
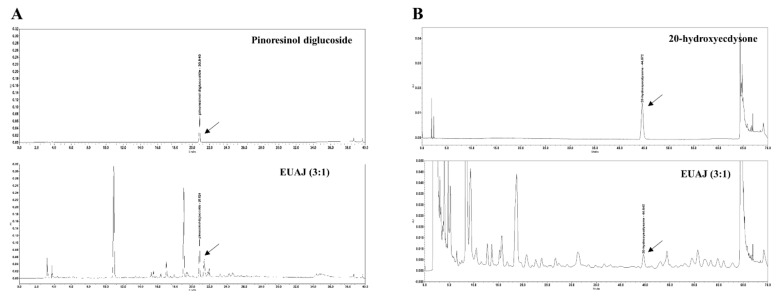
HPLC analysis of EUAJ (3:1). The contents of two ingredients in EUAJ (3:1) are shown using (**A**) pinoresinol diglucoside (PDG) and (**B**) 20-hydroxyecdysone (20HE) as standards.

**Figure 3 nutrients-14-03341-f003:**
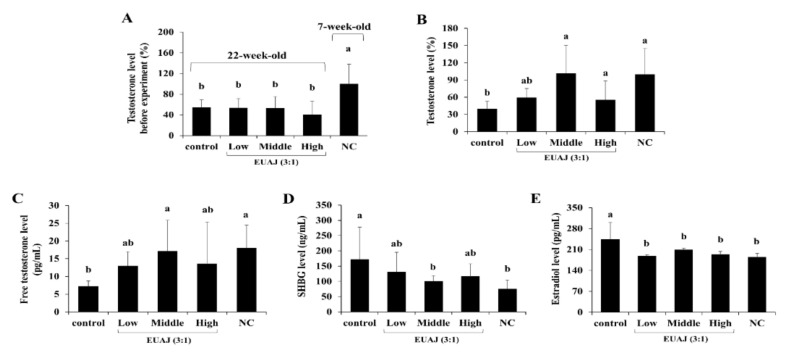
The effects of various concentration of EUAJ on total testosterone, free testosterone, SHBG and estradiol in serum of SD rats. Serum testosterone levels of SD rats were shown in (**A**) before administered by EUAJ (3:1). The level of (**B**) total testosterone, (**C**) free testosterone, (**D**) SHBG and (**E**) estradiol in serum of SD rats via oral gavage of various concentration of EUAJ (3:1) for 6 weeks. Values are presented as means ± SD. Different superscript letters show a significantly difference at *p* < 0.05, as determined by Duncan’s multiple range test. Control; saline, low; EUAJ (3:1) 20 mg/kg, middle; EUAJ (3:1) 40 mg/kg, high; EUAJ (3:1) 80 mg/kg, NC; saline.

**Figure 4 nutrients-14-03341-f004:**
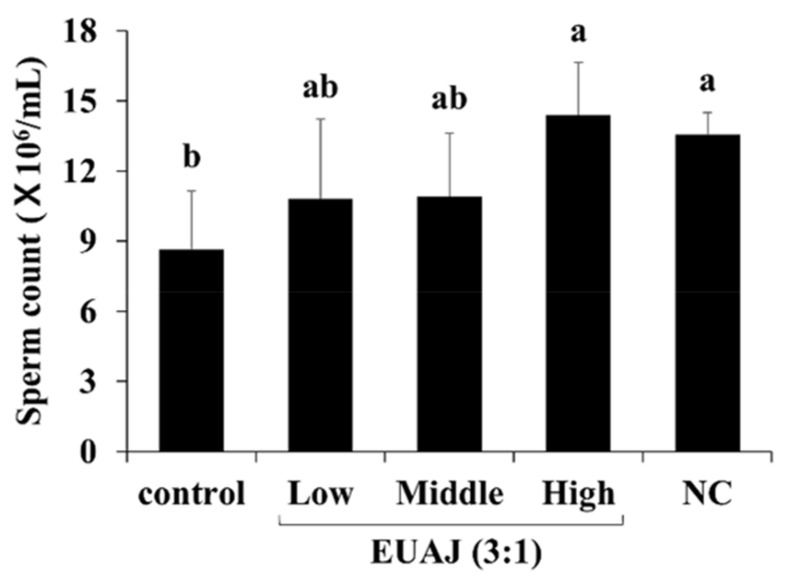
The effects of EUAJ (3:1) on sperm count in SD rats. Values are presented as means ± SD. Different superscript letters show a significantly difference at *p* < 0.05 as determined by Duncan’s multiple range test. Control; saline, low; EUAJ (3:1) 20 mg/kg, middle; EUAJ (3:1) 40 mg/kg, high; EUAJ (3:1) 80 mg/kg, NC; saline.

**Figure 5 nutrients-14-03341-f005:**
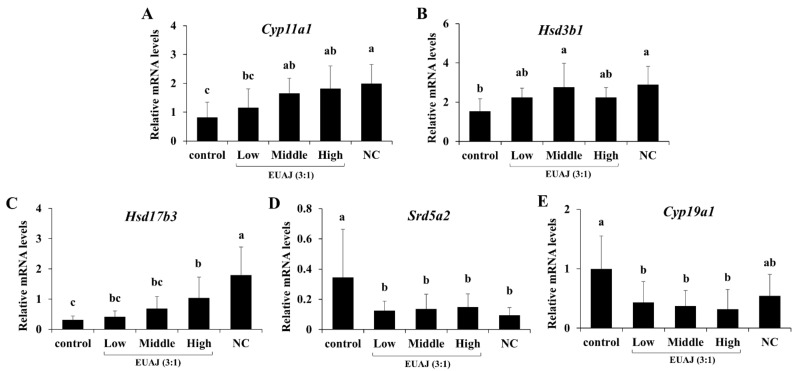
The effects of EUAJ (3:1) on mRNA expressions of steroidogenic enzymes involved in testosterone synthesis in testis of SD rats. mRNA expressions level of steroidogenic genes related to testosterone synthesis (**A**–**C**) and testosterone degradation (**D**,**E**) were measured by quantitative real-time PCR. Values are presented as means ± SD. Different superscript letters show a significantly difference at *p* < 0.05, as determined by Duncan’s multiple range test. Control; saline, low; EUAJ (3:1) 20 mg/kg, middle; EUAJ (3:1) 40 mg/kg, high; EUAJ (3:1) 80 mg/kg, NC; saline.

**Figure 6 nutrients-14-03341-f006:**
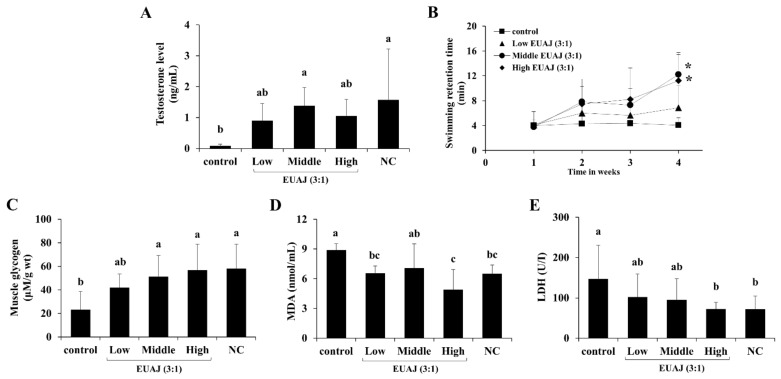
The effects of EUAJ (3:1) on testosterone level, swimming retention time, muscle glycogen, MDA and LDH in ICR mice. Serum testosterone levels of ICR mice administered by EUAJ (3:1) is shown in (**A**). Swimming retention time of ICR mice (**B**) was evaluated once a week for 4 weeks. (**C**) Muscle glycogen, (**D**) MDA and (**E**) LDH was measured after 30 min of swimming. Values are presented as means ± SD. Different superscript letters show a significantly difference at *p* < 0.05, as determined by Duncan’s multiple range test. Asterisk indicates *p* < 0.05 versus control. Control; saline, low; EUAJ (3:1) 20 mg/kg, middle; EUAJ (3:1) 40 mg/kg, high; EUAJ (3:1) 80 mg/kg, NC; saline.

**Figure 7 nutrients-14-03341-f007:**
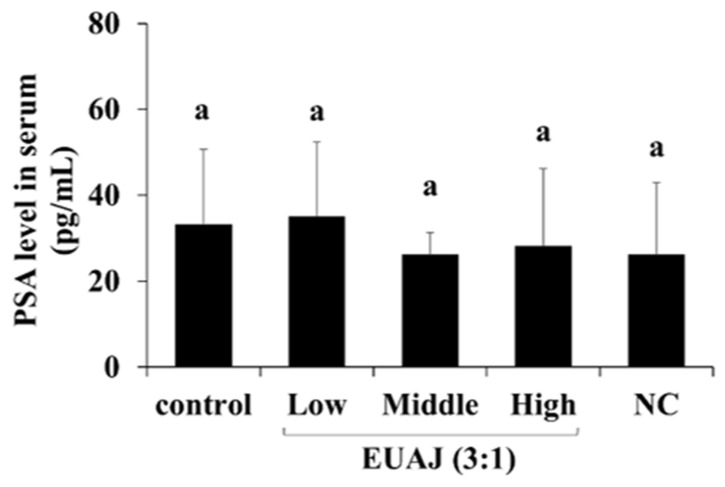
The effects of EUAJ (3:1) on serum PSA level in SD rats. PSA level were measured in serum of SD rats by administration of EUAJ (3:1). Values are presented as means ± SD. Different superscript letters show a significantly difference at *p* < 0.05, as determined by Duncan’s multiple range test. Control; saline, low; EUAJ (3:1) 20 mg/kg, middle; EUAJ (3:1) 40 mg/kg, high; EUAJ (3:1) 80 mg/kg, NC; saline.

**Figure 8 nutrients-14-03341-f008:**
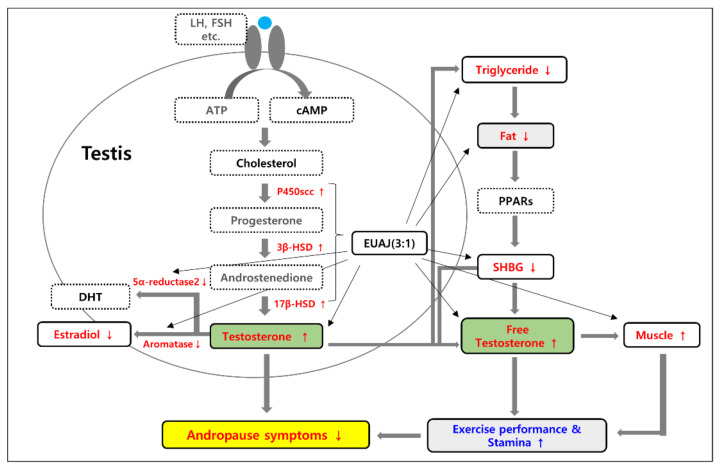
Schematic diagram showing the changes by EUAJ (3:1) treatment on the testicular steroidogenesis pathway and exercise performance in improving andropause symptoms.

**Table 1 nutrients-14-03341-t001:** The effects of EUAJ (3:1) on liver damage in SD rats.

Group	Liver Weight Ratio ^1^ (%)	ALT (KU)	AST (KU)
control	3.08 ± 0.24 ^a^	30.13 ± 3.44 ^a^	63.13 ± 14.84 ^a^
Low	2.99 ± 0.16 ^ab^	27.50 ± 6.68 ^ab^	67.88 ± 18.86 ^a^
Middle	2.86 ± 0.22 ^b^	27.00 ± 2.45 ^ab^	60.88 ± 10.95 ^a^
High	2.93 ± 0.20 ^ab^	24.63 ± 3.46 ^b^	56.38 ± 13.50 ^a^
NC	3.10 ± 0.13 ^a^	27.38 ± 2.26 ^ab^	68.13 ± 16.75 ^a^

^1^ Liver weight ratio (%): liver weight (g)/body weight (g) × 100. Values are presented as means ± SD. Different superscript letters show a significantly difference at *p* < 0.05, as determined by Duncan’s multiple range test. Control; saline, low; EUAJ (3:1) 20 mg/kg, middle; EUAJ (3:1) 40 mg/kg, high; EUAJ (3:1) 80 mg/kg, NC; saline.

**Table 2 nutrients-14-03341-t002:** The effects of EUAJ (3:1) on epididymal fat and muscle mass in SD rats.

Group	Epididymal Fat Weight (g)	Epididymal Fat Weight Ratio ^1^ (%)	Muscle Weight Ratio ^2^ (%)
Control	11.24 ± 0.99 ^a^	2.15 ± 0.20 ^a^	0.81 ± 0.02 ^c^
Low	10.10 ± 1.85 ^ab^	1.95 ± 0.33 ^ab^	0.86 ± 0.06b ^c^
Middle	8.68 ± 2.31 ^b^	1.65 ± 0.41 ^bc^	0.88 ± 0.03 ^ab^
High	9.88 ± 2.11 ^ab^	1.88 ± 0.40 ^ab^	0.89 ± 0.07 ^ab^
NC	5.83 ± 1.16 ^c^	1.47 ± 0.22 ^c^	0.93 ± 0.04 ^a^

^1^ Epididymal fat weight ratio (%); epididymal fat weight (g)/body weight (g) × 100. ^2^ Muscle weight ratio (%); total muscle weight (g)/body weight (g) × 100. Values are presented as means ± SD. Different superscript letters show a significantly difference at *p* < 0.05, as determined by Duncan’s multiple range test. Control; saline, low; EUAJ (3:1) 20 mg/kg, middle; EUAJ (3:1) 40 mg/kg, high; EUAJ (3:1) 80 mg/kg, NC; saline.

**Table 3 nutrients-14-03341-t003:** Serum lipid profiles in SD rats.

Group	TG (mg/dL)	Total-Cholesterol (mg/dL)	HDL-Cholesterol (mg/dL)	LDL-Cholesterol (mg/dL)
Control	194.13 ± 63.34 ^a^	118.63 ± 14.70 ^ab^	38.88 ± 8.49 ^a^	40.98 ± 18.26 ^a^
Low	193.75 ± 54.99 ^a^	129.63 ± 18.49 ^a^	42.88 ± 10.26 ^a^	47.38 ± 15.14 ^a^
Middle	128.25 ± 42.47 ^b^	121.63 ± 26.01 ^ab^	38.38 ± 8.40 ^a^	56.88 ± 25.39 ^a^
High	127.63 ± 44.34 ^b^	115.50 ± 13.85 ^ab^	36.38 ± 9.71 ^a^	52.53 ± 18.80 ^a^
NC	112.88 ± 42.12 ^b^	107.50 ± 22.06 ^b^	35.63 ± 11.49 ^a^	47.90 ± 12.70 ^a^

Values are presented as means ± SD. Different superscript letters show a significantly difference at *p* < 0.05, as determined by Duncan’s multiple range test. Control; saline, low; EUAJ (3:1) 20 mg/kg, middle; EUAJ (3:1) 40 mg/kg, high; EUAJ (3:1) 80 mg/kg, NC; saline.

## Data Availability

The data presented in this study are available upon request from the corresponding author.

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
