# Peer review of "The Effect of a Combination of Eucommia ulmoides and Achyranthes japonica on Alleviation of Testosterone Deficiency in Aged Rat Models"

_nutrients, 2022, doi:10.3390/nu14163341_

Round 1

Reviewer 1 Report

The Effect of KGC08EA (Eucomia ulmoides and Achyranthes japonica combination) on Alleviation of Testosterone Deficiency in Aged Animal Models

This is an interesting experiment and worth to be published.

Some remarks to improve the manuscript:

Title: ….in Aged Rat Models. (The difference in androgen production between man and rat is the lack of androgen synthesis in the adrenal gland of the rat – contrary to other animals - and therefore the effect of KGC08EA in rats is on testes.)

Line 42: 98% of serum testosterone is in the form of sex hormone-binding protein: 98% is the amount of protein-bound testosterone (SHBG plus other proteins like albumin).

Material and Methods

Please mention the time of the day when you administered KGC08EA and when (and from which vein) you collected blood.

Line 129 and 161: please give g-value instead of rpm.

You show mean and standard error. Did you check your results for Gaussian distribution?

Results

Fig. 1. I am confused: what is the difference between Testosterone and Total Testosterone? Why are the values in Fig. 1 B much lower?

Discussion:

Line 444: …..previous study (Literature missing)

Author Response

Dear Editor:

We wish to re-submit the manuscript titled “The Effect of a combination of Eucommia ulmoides and Achyranthes japonica on Alleviation of Testosterone Deficiency in Aged Rat Model.” The manuscript ID is nutrients-1842422.

We thank you and the reviewers for your thoughtful suggestions and insights. The manuscript has benefited from these insightful suggestions. I look forward to working with you and the reviewers to move this manuscript closer to publication in Nutrients.

The manuscript has been rechecked and the necessary changes have been made in accordance with the reviewers’ suggestions. The responses to all comments have been prepared attachment below. 

Thank you for your consideration. I look forward to hearing from you.

Point 1: Title: ….in Aged Rat Models. (The difference in androgen production between man and rat is the lack of androgen synthesis in the adrenal gland of the rat – contrary to other animals - and therefore the effect of KGC08EA in rats is on testes.)

Response 1: I deeply appreciate your detailed comments. The title was corrected in red letters. ‘…Deficiency in Aged Animal Models’ → ‘…Deficiency in Aged Rat Models’ (page 1, line 3)

Point 2: Line 42: 98% of serum testosterone is in the form of sex hormone-binding protein: 98% is the amount of protein-bound testosterone (SHBG plus other proteins like albumin).

Response 2: Thank you for your kind comment. On page 1, lines 42-43, We added the term ‘Sex hormone-binding globulin (SHBG)’ → ’protein-bound T such as sex hormone-binding globulin (SHBG)’

Point 3: Material and Methods

Please mention the time of the day when you administered KGC08EA and when (and from which vein) you collected blood.

Response 3: I deeply appreciate your detailed comments. The blood was drawn from the intraorbital vein blood collection (Page 3, line 100 ‘blood’ → ‘orbital blood’) The time for blood collection and oral administration is as follows in the manuscript on page 3, lines 135-136 → ‘All blood collection and oral administration were performed in the morning (9 am--11 am).’

Point 4: Line 129 and 161: please give g-value instead of rpm.

Response 4: The value for the centrifuging process has been included in the manuscript. ‘3000rpm’ → ‘1500g’ (Page 3, line 138)

Point 5: You show mean and standard error. Did you check your results for Gaussian distribution?

Response 5: Thank you for your kind comment. First, we apologize for this error. We changed the standard error to standard deviation. All significant differences are the same.

Point 6: Results

Fig. 1. I am confused: what is the difference between Testosterone and Total Testosterone? Why are the values in Fig. 1 B much lower?

Response 6: Thank you for your precise comment. First, we apologized for the lack of clarity regarding this. Testosterone and total testosterone are the same (We have unified them under the umbrella term “testosterone” to avoid confusion).

The reason for the difference in T values between A and B in Figure 1 arises from the use of two different kinds of testosterone ELISA measurement kit. Serum samples were treated in different ways before measurement, which is why testosterone levels between Fig1A and Fig2B are different. To avoid confusion and to facilitate an exact comparison between the two testosterone levels, we changed this to use as % testosterone levels instead of pg/mL. For transparent data disclosure in our research, we added the name of the testosterone measurement kit to the Material and Methods section (indicated by red text, page 3, line 143-144) and the original data were provided as supplementary data. Thank you again for the detailed comment.

- Serum without pretreatment:

Testosterone ELISA Kit (Abcam, Cambridge, UK) and a Rat Free Testosterone (F-TESTO) ELISA Kit (Cusabio, Wuhan, China)

- Serum with pretreatment:

DetectX® Testosterone ELISA Kits (Arbor assays, MI, United States)

Prior to assays, serum was extracted according to the manufacturer’s instructions to eliminate contaminants, such as bulk proteins and lipids. Ethyl acetate was added to serum at a 5:1 (v/v) ratio and mixed by vortexing. After 5 minutes of incubation, the separated solvent layer was once frozen and poured into sterile tube. The steps were repeated twice for maximum extraction efficiency. The solvents were combined and kept desiccated at -20°C. The samples were redissolved at room temperature for reconstitution.

Point 7: Discussion:

Line 444: …..previous study (Literature missing)

Response 7: Thank you for your kind comment. The reference have been added in the manuscript (page 13, line 465-466). → ‘Similar to a previous study [51]’

Reviewer 2 Report

The current ms need major revision, and I list some correction as follow,

1. For the title of ms, KGC08EA is not appropriate to appear in paper title, because this is belong a commercial name, there will be a big concerns that the journal nutrients is endorsement the pharmaceutical functions or efficacy, especially for the human-being healthy products, the complete name of two herb is enough, The Effect of Combination of Eucomia ulmoides and Achyranthes japonica on Alleviation of Testosterone Deficiency in Aged Animal Models. Revision also need to be done in the whole text, an academic name in short such as ‘EUAJ’ is more suitable.

2. Page 3, line 103, There is only seven groups in the experiment design.

3. Page 3, For three experiment design, please list all parameter that will be determined in the end of description of the paragraph.

4. Page 3, line137. The correct name “serum-bound SHBG “  

5. Page 5. The combination of EUAJ were 3:1, and the HPLC analysis showed the main compounds is 20.09mg/kg and 0.41mg/kg, after combination (75% vs 25%) of EUAJ definitely need to show the final results for both compounds by HPLC quantitative analysis information. However, this ms need complete this analysis to make sure the final concentration of main compounds before ms is accepted.  

6. Page 6. Line 241, orally injected is not the correct name for dosing, oral gavage will be the way for administration, please double confirm the whole text description.

7. Page 10, line 333-339 move under the figure results.

8. Discussion need to be more explanation and revision, for example the possible mechanism of current results or combination or individual administrated will have some different results on spermatogenesis, and more literature will be cited, theres are lots of Chinese herbs functions of Eucomia ulmoides published.

9. Page 12, line 417-429. How to explain the current results which 3:1 components or compounds give the final findings.

10. Page 8, Table 1. The results of ALT decrease in high dosage groups, what is the mechanism please description in the discussion section. Muscle weight ratio in table 2 is also need to more discussion an explanation.

11. Page 9 Table 3. Serum lipid profiles in SD rats is suggest.

12. Figure 3. B and D, middle dosage give the highest level of testosterone, more discussion or cross-talk and what is mechanism of AUAJ works on modulatory effects on steroidogenesis pathways which observed in the Figure 5.

Author Response

Dear Editor:

We wish to re-submit the manuscript titled “The Effect of a combination of Eucommia ulmoides and Achyranthes japonica on Alleviation of Testosterone Deficiency in Aged Rat Model.” The manuscript ID is nutrients-1842422.

We thank you and the reviewers for your thoughtful suggestions and insights. The manuscript has benefited from these insightful suggestions. I look forward to working with you and the reviewers to move this manuscript closer to publication in Nutrients.

The manuscript has been rechecked and the necessary changes have been made in accordance with the reviewers’ suggestions. The responses to all comments have been prepared below. 

Thank you for your consideration. I look forward to hearing from you.

Point 1: For the title of ms, KGC08EA is not appropriate to appear in paper title, because this is belong a commercial name, there will be a big concerns that the journal nutrients is endorsement the pharmaceutical functions or efficacy, especially for the human-being healthy products, the complete name of two herb is enough, The Effect of Combination of Eucommia ulmoides and Achyranthes japonica on Alleviation of Testosterone Deficiency in Aged Animal Models. Revision also need to be done in the whole text, an academic name in short such as ‘EUAJ’ is more suitable.

Response 1: Thank you for your kind comment. We fully understand your concern and agree with your advice. Following the referee’s suggestion, we changed KGC08EA to ‘Combination of Eucommia ulmoides and Achyranthes japonica’ in the title.

It should be noted that KGC08EA is a sample code name. However, to avoid misunderstanding, we agree that use the phrase ‘Combination of Eucommia ulmoides and Achyranthes japonica’ instead of KGC08EA in the title. Also, we changed the term ‘KGC08EA’ to ‘EUAJ’ throughout the manuscript except in the ‘Material and Method’ and ‘Abstract’ sections, because we screened many combinations of the two extracts, and there was a need to indicate the exact sample code. Please let me know if any further corrections are required. Thank you so much for your kind comments.

Point 2: Page 3, line 103, There is only seven groups in the experiment design.

Response 2: I really appreciate your detailed comments. We changed ‘8 groups’ → ‘7 groups’ in the manuscript (page 3, line 104). In addition, other typographical errors in the manuscript were corrected, and awkward sentences were also amended.

Page 2, line 63) ‘D. japonica’ → ‘A. japonica

Page 6, line 226, 230) ‘two active ingredients’ → ‘two ingredients’

Page 12, line 448-450) We corrected the following awkward sentences due to repeated words (underlined) below.

‘Under fatigue conditions, MDA, the main product of membrane lipid peroxidation directly contributes to the pathophysiology of fatigue [38]. Under fatigue conditions, MDA, the main product of membrane lipid peroxidation by free radicals, is increased [36].’

→ ‘Under fatigue conditions, MDA, the main product of membrane lipid peroxidation by free radicals, increases and directly contributes to the pathophysiology of fatigue [48,50].’

Point 3: Page 3, For three experiment design, please list all parameter that will be determined in the end of description of the paragraph.

Response 3: The following sentences were added to the paragraphs describing each experiment design.

Page 3, line 110-113) ‘Six weeks later, animal blood was collected again to measure serum testosterone concentration. The mixture ratio of aj and eu, which demonstrated the greatest increase in testosterone levels, was selected and named EUAJ.’

Page 3, line 120-123) ‘Thereafter, T levels, free T levels, ALT, AST, lipid profile, SHBG, estradiol levels, and PSA levels in SD rat serum were investigated. Steroidogenesis gene mRNA expression was also confirmed in animal testis tissue. Finally, epididymal fat ratio, muscle weight ratio, and sperm count were investigated.’

Page 3, line 132-135) ‘Then, T levels, malondialdehyde (MDA), and lactate dehydrogenase (LDH) were confirmed in ICR mouse serum. In addition, the presence of muscle glycogen was confirmed in animal muscle tissue, and swimming retention time was investigated for a total of 4 weeks.’

Point 4: Page 3, line137. The correct name “serum-bound SHBG “   

Response 4: Thank you for your kind comment. The name was corrected as indicated by red-colored text in the manuscript. The term ‘serum-binding SHBG’ → ‘serum-bound SHBG’ (Page 3, line 147)

Point 5: Page 5. The combination of EUAJ were 3:1, and the HPLC analysis showed the main compounds is 20.09mg/kg and 0.41mg/kg, after combination (75% vs 25%) of EUAJ definitely need to show the final results for both compounds by HPLC quantitative analysis information. However, this ms need complete this analysis to make sure the final concentration of main compounds before ms is accepted.

Response 5: First, we apologize for the inaccurate description which may have caused confusion. EUAJ was extracted together and then subjected to HPLC. The Material and Methods was modified as follows.

Page 2, line 63-66) ‘The bark of E. ulmoides and radix of A. japonica were purchased from Kyungil Medicinal Herbs (Geumsan, Korea), and subsequently mixed (E. ulmoides : A. japonica = 3:1, KGC08EA) and extracted twice with 30% ethanol. The extracts were filtered, concentrated under vacuum, and spray-dried (with yields of 11.0% and 45%, respectively).’

Point 6: Page 6. Line 241, orally injected is not the correct name for dosing, oral gavage will be the way for administration, please double confirm the whole text description.

Response 6: Thank you for your kind comment. The term was corrected in the manuscript, as indicated by the red-colored text. The term ‘orally injected’ → ‘oral gavage of’ (Page 6, line 249)

Point 7: Page 10, line 333-339 move under the figure results.

Response 7: Thank you for your thoughtful comment. Following the referee’s suggestion, the figure legend text has been moved under the relevant figure results.

Point 8: Discussion need to be more explanation and revision, for example the possible mechanism of current results or combination or individual administrated will have some different results on spermatogenesis, and more literature will be cited, theres are lots of Chinese herbs functions of Eucommia ulmoides published.

Response 8: I have added the following text to the Discussion section below, thank you for the suggestion. (page 12, line 425-431)

→ ‘Eucommia ulmoides has been used as a traditional medicinal plant supplement to prevent male infertility. According to a recent study, the Chinese prescription, Duzhong Butiansu (DZBTS), which contains E. ulmoides Oliv. down-regulates heat stress, thereby reducing oxidative stress and improving spermatogenesis [38]. In addition, aucubin, an iridoid glucoside extracted from E. ulmoides, protected spermatogenesis by preventing apoptosis of sertoli cells by upregulation of Nrf2 and induction of antioxidant responses from triptolide-induced testicular damage [39].’

Point 9: Page 12, line 417-429. How to explain the current results which 3:1 components or compounds give the final findings.

Response 9: Figure 8 was modified to show the effect of EUAJ (3:1) components in our final results. (Page 13) EUAJ increased the level of testosterone through upregulation of the steroidogenesis pathway gene expressions and ultimately improved the symptoms of andropause.

Point 10: Page 8, Table 1. The results of ALT decrease in high dosage groups, what is the mechanism please description in the discussion section. Muscle weight ratio in table 2 is also need to more discussion an explanation.

Response 10: EU (Eucommia ulmoides) has been used as a traditional medicinal remedy for nourishing liver, kidney and the musculoskeletal system in China. A recent study reported that EU decreased serum ALT and AST in rats with CCl4-induced hepatic damaged [1,2]. It is also reported that pinoresinol from EU, the plant lignan, produced through phenylpropanoid pathway with pinoresinol glucoside (PDG) [3], and which is suggested to have a common bioactivity profile with PDG, also protects the liver from CCl4-induced hepatic damage and reduces serum ALT and AST [4]. In this experiment, ALT and AST tended to decrease by EUAJ (3:1) treatment, and ALT significantly decreased at high concentrations. This is the result of the hepatoprotective effect of EU via plant lignans such as pinoresinol and PDG. (The sentence is included in the Discussion section on page 11, line 404-406.)

Furthermore, regarding the effect of EUAJ on muscle mass, data from the five muscle regions (Gastrocnemius, Tibialis anterior, Soleus, EDL, Plantaris) used to calculate the muscle weight ratio below are attached.

Figure 1. Muscle weight ratio of gastrocnemius, tibialis anterior, soleus, EDL, and plantaris from SD rat.

Testosterone is strongly associated with muscle mass and muscle protein synthesis [5–7]. Testosterone is a potent anabolic in skeletal muscle, regulating protein metabolism [8–10]. The reason EUAJ could affect muscle mass seems to be the increase in testosterone caused by EUAJ. Although only the middle concentration of EUAJ (3:1) showed a significant increase in testosterone, it was confirmed that the muscle weight ratio increased at both middle and high concentrations of EUAJ (3:1). (The sentence is attached to the Discussion section on page 12, lines 432-435.)

Point 11: Page 9 Table 3. Serum lipid profiles in SD rats is suggest.

Response 11: Thank you for your kind comment. The suggesting term was added to the manuscript, as indicated by the red-colored text in the manuscript. (Page 9, line 316) ‘Lipid profiles in serum of SD rats’ → ‘Serum lipid profiles in SD rats’

Point 12: Figure 3. B and D, middle dosage give the highest level of testosterone, more discussion or cross-talk and what is a mechanism of AUAJ works on modulatory effects on steroidogenesis pathways which observed in  Figure 5.

Response 12: In the steroidogenesis pathway, the preference for either Δ4 or Δ5 pathways in species differences is due to the relative substrate affinity of the CYP17 enzyme. In the rat, CYP17 shows a preference for the Δ4 pathway, containing the intermediates progesterone and 17α-hydroxy-progesterone (figure 2, below) [11]. In this study, Hsd3b1 showed a significant increase at the middle concentration of EUAJ (3:1), and it means that the role of Hsd3b1 in the pathway from Δ5 to Δ4 might be relatively increased at the middle concentration. This result led to a predominant increase in testosterone at the middle concentration of EUAJ (3:1). (The sentence is attached to the Discussion section on page 11, lines 389-395.)

 Figure 2. Main components of the steroidogenic pathway in the human (left) and murine (right) fetal Leydig cell [11].

References

  1. Lee, H.Y.; Lee, G.H.; Yoon, Y.; Chae, H.J. Rhus Verniciflua and Eucommia Ulmoides Protects Against High-Fat Diet-Induced Hepatic Steatosis by Enhancing Anti-Oxidation and AMPK Activation. Am J Chin Med (Gard City N Y) 2019, 47, 1253–1270, doi:10.1142/S0192415X19500642.
  2. Jin, C.F.; Li, B.; Lin, S.M.; Yadav, R.K.; Kim, H.R.; Chae, H.J. Mechanism of the Inhibitory Effects of Eucommia Ulmoides Oliv. Cortex Extracts (EUCE) in the CCl 4 -Induced Acute Liver Lipid Accumulation in Rats. Int J Endocrinol 2013, 2013, doi:10.1155/2013/751854.
  3. Pu, Y.; Cai, Y.; Zhang, Q.; Hou, T.; Zhang, T.; Zhang, T.; Wang, B. Comparison of Pinoresinol and Its Diglucoside on Their ADME Properties and Vasorelaxant Effects on Phenylephrine-Induced Model. Front Pharmacol 2021, 12, doi:10.3389/FPHAR.2021.695530.
  4. Kim, H.Y.; Kim, J.K.; Choi, J.H.; Jung, J.Y.; Oh, W.Y.; Kim, D.C.; Lee, H.S.; Kim, Y.S.; Kang, S.S.; Lee, S.H.; et al. Hepatoprotective Effect of Pinoresinol on Carbon Tetrachloride-Induced Hepatic Damage in Mice. J Pharmacol Sci 2010, 112, 105–112, doi:10.1254/JPHS.09234FP.
  5. Griggs, R.C.; Kingston, W.; Jozefowicz, R.F.; Herr, B.E.; Forbes, G.; Halliday, D. Effect of Testosterone on Muscle Mass and Muscle Protein Synthesis. J Appl Physiol (1985) 1989, 66, 498–503, doi:10.1152/JAPPL.1989.66.1.498.
  6. Bhasin, S.; Woodhouse, L.; Storer, T.W. Proof of the Effect of Testosterone on Skeletal Muscle. J Endocrinol 2001, 170, 27–38, doi:10.1677/JOE.0.1700027.
  7. Urban, R.J.; Bodenburg, Y.H.; Gilkison, C.; Foxworth, J.; Coggan, A.R.; Wolfe, R.R.; Ferrando, A. Testosterone Administration to Elderly Men Increases Skeletal Muscle Strength and Protein Synthesis. Am J Physiol 1995, 269, doi:10.1152/AJPENDO.1995.269.5.E820.
  8. Singh, R.; Artaza, J.N.; Taylor, W.E.; Gonzalez-Cadavid, N.F.; Bhasin, S. Androgens Stimulate Myogenic Differentiation and Inhibit Adipogenesis in C3H 10T1/2 Pluripotent Cells through an Androgen Receptor-Mediated Pathway. Endocrinology 2003, 144, 5081–5088, doi:10.1210/EN.2003-0741.
  9. Sinha-Hikim, I.; Roth, S.M.; Lee, M.I.; Bhasin, S. Testosterone-Induced Muscle Hypertrophy Is Associated with an Increase in Satellite Cell Number in Healthy, Young Men. Am J Physiol Endocrinol Metab 2003, 285, doi:10.1152/AJPENDO.00370.2002.
  10. Dubois, V.; Laurent, M.; Boonen, S.; Vanderschueren, D.; Claessens, F. Androgens and Skeletal Muscle: Cellular and Molecular Action Mechanisms Underlying the Anabolic Actions. Cell Mol Life Sci 2012, 69, 1651–1667, doi:10.1007/S00018-011-0883-3.
  11. Scott, H.M.; Mason, J.I.; Sharpe, R.M. Steroidogenesis in the Fetal Testis and Its Susceptibility to Disruption by Exogenous Compounds. Endocrine Reviews 2009, 30, 883–925, doi:10.1210/ER.2009-0016.

Round 2

Reviewer 2 Report

All revisions meet the suggested, correct mis-wording and discussion section already added.